# Carbamazepine Restores Neuronal Signaling, Protein Synthesis, and Cognitive Function in a Mouse Model of Fragile X Syndrome

**DOI:** 10.3390/ijms21239327

**Published:** 2020-12-07

**Authors:** Qi Ding, Fan Zhang, Yue Feng, Hongbing Wang

**Affiliations:** 1Department of Physiology, Michigan State University, East Lansing, MI 48824, USA; dingq@msu.edu (Q.D.); zhang124fan356@163.com (F.Z.); 2Department of Pharmacology and Chemical Biology, Emory University School of Medicine, Atlanta, GA 30322, USA; yfeng@emory.edu; 3Neuroscience Program, Michigan State University, East Lansing, MI 48824, USA

**Keywords:** fragile X syndrome, therapeutics, mouse model, carbamazepine, type 1 adenylyl cyclase, phosphodiesterase, cAMP, ERK½, PI3 kinase, protein translation, intellectual disability, learning and memory, drug repurposing

## Abstract

Fragile X syndrome (FXS) is a leading genetic disorder of intellectual disability caused by the loss of the functional fragile X mental retardation protein (FMRP). To date, there is no efficacious mechanism-based medication for FXS. With regard to potential disease mechanisms in FXS, it is widely accepted that the lack of FMRP causes elevated protein synthesis and deregulation of neuronal signaling. Abnormal enhancement of the ERK½ (extracellular signal-regulated kinase ½) and PI3K-Akt (Phosphoinositide 3 kinase-protein kinase B) signaling pathways has been identified in both FXS patients and FXS mouse models. In this study, we show that carbamazepine, which is an FDA-approved drug and has been mainly used to treat seizure and neuropathic pain, corrects cognitive deficits including passive avoidance and object location memory in FXS mice. Carbamazepine also rescues hyper locomotion and social deficits. At the cellular level, carbamazepine dampens the elevated level of ERK½ and Akt signaling as well as protein synthesis in FXS mouse neurons. Together, these results advocate repurposing carbamazepine for FXS treatment.

## 1. Introduction

Fragile X syndrome (FXS) is a leading cause of inherited intellectual disability and autism. Genetic studies have demonstrated that loss of function mutations in the fragile X mental retardation 1 gene (*FMR1*) that encodes the fragile X mental retardation protein (FMRP) results in FXS [1,2]. Although the disease mechanism remains largely elusive, multiple lines of data demonstrate that FMRP interacts with both mRNAs and protein translation machinery [1,2,3]. It has been found that, except for a few specific targets [4], FMRP deficiency causes an overall enhancement of protein synthesis [3]. There are also lines of clinical and preclinical evidence suggesting altered neuronal signaling in FXS. Previous studies have found that activity of the ERK½ signaling pathway is abnormally enhanced in FXS mouse model (i.e., *Fmr1* knockout mouse) and human patient samples [5,6,7]. With regard to understanding how alteration of FMRP-regulated translation is linked to elevated ERK½ activity, it is found that FMRP binds *Adcy1* (type 1 adenylyl cyclase) mRNA [8,9], and ADCY1 protein level is elevated in *Fmr1* knockout (KO) mice [10]. Consistent with the finding that enhancement of ADCY1 expression causes elevated ERK½ activity [11] and a panel of FXS-associated behavior (e.g., hyperlocomotion and social deficits) [12], genetic reduction of *Adcy1* attenuates the elevated ERK½ signaling and the aforementioned behavioral abnormalities in *Fmr1* KO mice [10]. Intriguingly, genetic reduction of *Adcy1* also restores Akt signaling [10], the activity of which is elevated in both human patient and FXS mouse samples [7,13,14]. These observations suggest that aberrant hyperfunction of ADCY1 in FXS is a potential therapeutic target, through which the pathological overactivation of both ERK½ and Akt signaling pathways can be corrected. However, the potential therapeutic value in clinical application requires examination with practical pharmacological intervention.

Carbamazepine is an FDA-approved drug and is currently used for the treatment of seizure and neuropathic pain. Because carbamazepine shows pharmacological activity against ADCY1 [15], we reason that it may elicit therapeutic effects on FXS-associated pathological phenotypes. In contrast to ADCY, which contributes to cAMP production, PDE (phosphodiesterase) possesses enzymatic activity to reduce cAMP level by converting it to AMP. Notably, it is reported that FMRP also regulates the translation of certain PDE subtype (e.g., PDE2A) and the PDE2A protein level is elevated in *Fmr1* KO mice [16]. Thus, it is intriguing and seemingly possible that FMRP deficiency could concurrently affect signaling molecules with counteracting enzymatic activity.

To better understand this complication and the therapeutic potential through inhibiting ADCY or PDE2, in this study, we compared ADCY1 and PDE2A expression in wild type (WT) and *Fmr1* KO mice. Furthermore, we demonstrated that inhibition of ADCY1 by carbamazepine can correct abnormal ERK½ and Akt signaling, protein synthesis, and the core symptoms of cognitive deficits in a mouse model of FXS. However, PDE2A inhibitors can only attenuate the elevated Akt activity but have no effect on correcting the abnormal ERK½ activity and protein synthesis. These results provide the first evidence demonstrating differential pathological contributions of enzymatic components in cAMP homeostasis in FXS.

## 2. Results

### 2.1. ADCY1 Is Expressed in Neurons and Elevated in the Dorsal Hippocampus of Fmr1 KO Mice

*Adcy1* is predominantly expressed in the central nervous system (CNS) [17,18], making it an attractive drug target for the treatment of neurological disorders. Here, we further found that ADCY1 is expressed in neurons but not glial cells. The enrichment of neurons or glial cells in our primary cultures was confirmed by immunoblots using antibodies against neuronal and glial-specific marker proteins (Figure 1). Clearly, ADCY1 expression is restricted in neurons enriched of synapsin I (Figure 1a). Extracts from glial culture show expression of glial fibrillary acidic protein (GFAP) but not ADCY1 (Figure 1a). Interestingly, while ERK½ show robust expression in both cell types, Akt is predominately expressed in neurons, far exceeding that in glial cells (Figure 1a).

FMRP interacts with *Adcy1* mRNA and suppress its translation [8,9,10]. We confirmed that ADCY1 protein level is elevated in the hippocampus of *Fmr1* KO mice (Figure 1b). Interestingly, the elevated ADCY1 expression is brain region specific. It is preferentially elevated in the dorsal hippocampus but not in ventral hippocampus and prefrontal cortex of *Fmr1* KO mice (Figure 1c).

### 2.2. Carbamazepine Restores Hippocampus-Dependent Memory in Fmr1 KO Mice

Recapitulating the intellectual disability in human FXS patients, *Fmr1* KO mice display deficits in various learning and memory tasks including the hippocampus-dependent passive avoidance memory [19,20]. We administered carbamazepine through i.p. injection, followed by passive avoidance training. The carbamazepine- and vehicle-treated WT and *Fmr1* KO mice showed comparable crossover latency during training (genotype effect: *F_1, 39_* = 3.404, *p* = 0.073; treatment effect: *F_1, 39_* = 1.459, *p* = 0.234; genotype X treatment interaction: *F_1, 39_* = 1.501, *p* = 0.228) (Figure 2a). When tested 24 h later, the vehicle-treated *Fmr1* KO mice showed significantly shorter crossover latency than the vehicle-treated WT mice (*p* = 0.009, Fisher exact test). Carbamazepine improved the memory in *Fmr1* KO mice to the WT level (*p* = 0.495, Fisher exact test) (Figure 2a).

Next, we tested the effects of carbamazepine on object location memory (Figure 2(b1)), which mainly depends on dorsal hippocampus [21,22]. We found that *Fmr1* KO mice are impaired to establish object location memory. While all groups of animals showed comparable preference for objects in location A and B during training (Figure 2(b2)) (genotype effect: *F_1, 62_* = 0.000, *p* = 1.000; treatment effect: *F_1, 62_* = 0.000, *p* = 1.000; genotype X treatment interaction: *F_1, 62_* = 3.691, *p* = 0.059), the vehicle-treated WT but not *Fmr1* KO mice showed preference for the object in the new location (i.e., location C) over the object in the old location (i.e., location A) during testing (Figure 2(b3)) (WT: *p* = 0.018; *Fmr1* KO: *p* = 0.991). The carbamazepine-treated WT and *Fmr1* KO mice both showed preference for the new location (Figure 2(b3)) (genotype X location: *F_1, 62_* = 6.240, *p* = 0.015; treatment X location: *F_1, 62_* = 11.310, *p* = 0.001; genotype X treatment X location: *F_1, 62_* = 0.003, *p* = 0.955; WT: *p* < 0.001; *Fmr1* KO: *p* = 0.001). However, a single injection of carbamazepine did not improve object discrimination index in *Fmr1* KO mice (Figure 2(b4)) (genotype effect: *F_1, 31_* = 3.12, *p* = 0.087; treatment effect: *F_1, 31_* = 5.655, *p* = 0.024; genotype X treatment interaction: *F_1, 31_* = 0.002, *p* = 0.968). Interestingly, *Fmr1* KO mice receiving repeated daily administration of carbamazepine for 8 days displayed normal behavior during training (Figure 2(c1)) (genotype effect: *F_1, 62_* = 0.000, *p* = 1.000; treatment effect: *F_1, 62_* = 0.000, *p* = 1.000; genotype X treatment interaction: *F_1, 62_* = 0.003, *p* = 0.959). They showed improved object location memory (Figure 2(c2)) (vehicle: *p* = 0.299; carbamazepine: *p* < 0.001) and object discrimination index during testing (Figure 2(c3)) (genotype effect: *F_1, 31_* = 0.396, *p* = 0.534; treatment effect: *F_1, 31_* = 0.846, *p* = 0.365; genotype X treatment interaction: *F_1, 31_* = 9.819, *p* = 0.004).

### 2.3. Effects of Carbamazepine on Non-Cognitive Behavioral Abnormalities in Fmr1 KO Mice

Besides intellectual disability, other core FXS behavioral symptoms include hyperactivity and social deficits. We examined the effects of carbamazepine on locomotion activity. One hour after carbamazepine or vehicle injection, mice were placed in an open field arena. Consistent with previous studies [2,19], vehicle-treated *Fmr1* KO mice showed hyper-locomotion than WT mice in the whole arena (Figure 3(a1,a1′); genotype effect: *F_1, 52_* = 23.814, *p* < 0.001; treatment effect: *F_1, 52_* = 0.222, *p* = 0.640; genotype X treatment interaction: *F_1, 52_* = 6.147, *p* = 0.016) and the center area of the arena (Figure 3(a2,a2′); genotype effect: *F_1, 52_* = 22.069, *p* < 0.001; treatment effect: *F_1, 52_* = 3.339, *p* = 0.073; genotype X treatment interaction: *F_1, 52_* = 9.449, *p* = 0.003) (Figure 3(a3,a3′); genotype effect: *F_1, 52_* = 25.098, *p* < 0.001; treatment effect: *F_1, 52_* = 0.286, *p* = 0.595; genotype X treatment interaction: *F_1, 52_* = 9.113, *p* = 0.004). Carbamazepine normalized whole arena locomotion in *Fmr1* KO mice to the level observed in WT mice (Figure 3(a1), *p* = 0.089 between the treated *Fmr1* KO and WT mice). Carbamazepine also normalized hyper-locomotion in the center area of the open field arena in *Fmr1* KO mice (Figure 3(a2); *p* = 0.246 between the treated *Fmr1* KO and WT mice) (Figure 3(a3), *p* = 0.156 between the treated *Fmr1* KO and WT mice).

In the 3-chamber social interaction test, all groups spent similar total time in the chamber that holds the stranger mouse (Appendix A) (Figure 3(b1); genotype effect: *F_1, 30_* = 1.188, *p* = 0.284; treatment effect: *F_1, 30_* = 0.002, *p* = 0.968; genotype X treatment interaction: *F_1, 30_* = 2.861, *p* = 0.101). Notably, *Fmr1* KO mice spent less time in direct interaction with the stranger mouse than WT mice (Appendix A, Figure 3(b2)). While a single acute treatment with carbamazepine had no significant effect (Appendix A), repeated daily administration for 8 days improved social interaction in *Fmr1* KO mice (Figure 3(b2)). It is interesting to note that repeated carbamazepine also increased social interaction in WT mice (Figure 3(b2)), implicating that social behavior in WT mice can also be improved. Importantly, following repeated carbamazepine administration, WT and *Fmr1* KO mice showed comparable social interaction (Figure 3(b2)); genotype effect: *F_1, 30_* = 19.521, *p* < 0.001; treatment effect: *F_1, 30_* = 22.184, *p* < 0.001; genotype X treatment interaction: *F_1, 30_* = 2.620, *p* = 0.116).

Our previous studies found that *Fmr1* KO mice made more transitional moves between two chambers in the light/dark test, reflecting hyperactivity and repetitive behavior [10,19]. For this specific behavioral abnormality, neither single (Appendix A) nor repeated treatment (Appendix A) with carbamazepine exerted significant therapeutic effect. Interestingly, repeated carbamazepine increased total time spent in the light chamber in both WT and *Fmr1* KO mice (Appendix A).

### 2.4. Carbamazepine Dampens the Elevated Protein Synthesis in Fmr1 KO Neurons

Consistent with previous reports, we found that protein synthesis is elevated in *Fmr1* KO hippocampal neurons (Figure 4(a1,a2); genotype effect: *F_1, 16_* = 12.898, *p* = 0.002). Carbamazepine dampened protein synthesis in *Fmr1* KO neurons in a dose-dependent manner without affecting wild type neurons (Figure 4(a1,a2); treatment effect: *F_3, 16_* = 3.757, *p* = 0.032; genotype X treatment interaction: *F_3, 16_* = 5.081, *p* = 0.012).

### 2.5. Carbamazepine Suppresses Both ERK½ and Akt Activity in Neurons

The elevated ERK½ and Akt activity, which may contribute to the exaggerated protein synthesis in FXS neurons, has been previously observed in brain tissues of *Fmr1* KO mice. As the brain tissue consists of both neuron and non-neuronal cell types (e.g., glial cells), it is not clear whether the abnormal activity of ERK½ and Akt is neuron specific or due to changes in non-neuronal cells. Here, we used neuron-enriched primary cultures to determine whether there is neuron specific alteration of ERK½ and Akt and examine the therapeutic effect of carbamazepine.

As reflected by the levels of phosphorylated ERK½ and Akt (i.e., pERK½ and pAkt, respectively), the activity of ERK½ (Figure 4(b1,b2); genotype effect: *F_1, 16_* = 8.788, *p* = 0.009) and Akt (Figure 4(c1,c2); genotype effect: *F_1, 16_* = 57.920, *p* < 0.001) are elevated in *Fmr1* KO hippocampal neurons.

Carbamazepine dose-dependently decreases the level of pERK½ (Figure 4(b1,b2); treatment effect: *F_3, 16_* = 129.537, *p* < 0.001; genotype X treatment interaction: *F_3, 16_* = 21.401, *p* < 0.001) and pAkt (Figure 4(c1,c2); treatment effect: *F_3, 16_* = 14.840, *p* < 0.001; genotype X treatment interaction: *F_3, 16_* = 3.233, *p* = 0.050) in both WT and *Fmr1* KO neurons. The abnormally elevated pERK½ and pAkt in *Fmr1* KO neurons were more sensitive to carbamazepine treatment than wild type neurons. Comparing to wild type neurons, pERK½ and pAkt in FXS neurons were effectively suppressed by lower concentrations of carbamazepine. While 10 μM carbamazepine was sufficient to dampen pERK½ and pAkt in *Fmr1* KO neuron, higher concentration (100 μM) was required for WT neurons (Figure 4(b1–b3,c1–c3)).

The level of total ERK½ (Figure 4(b1,b3); genotype effect: *F_1, 16_* = 0.027, *p* = 0.871; treatment effect: *F_3, 16_* = 0.760, *p* = 0.533; genotype X treatment interaction: *F_3, 16_* = 0.303, *p* = 0.308) and Akt (Figure 4(c1,c3); genotype effect: *F_1, 16_* = 0.000, *p* = 0.999; treatment effect: *F_3, 16_* = 0.089, *p* = 0.965; genotype X treatment interaction: *F_3, 16_* = 0.293, *p* = 0.830) was not affected by genotype and carbamazepine treatment.

### 2.6. Pharmacological Inhibition of PDE2 and Activation of ADCY Do Not Correct the Elevated Protein Synthesis in FXS Mouse Neurons

ADCY and PDE have counteracting enzymatic action on cAMP metabolism and may have different effects on cAMP-regulated signaling molecules such as ERK½ [11,23]. With regard to the consideration of targeting ADCY1-ERK½ for potential therapeutic strategy, one complication is that PDE2A was also identified as an FMRP target [16]. We found that PDE2A is also primarily expressed in neurons but not glial cells (Figure 5a). Interestingly, PDE2A protein is not altered in adult *Fmr1* KO hippocampus (Figure 5b). This is consistent with the idea that PDE2A may be affected only in juvenile but not adult FXS mice [16,24].

In contrast to the effects of carbamazepine, BAY607550, which has inhibitory activity against PDE2 [25], did not show significant effect on protein translation (Figure 5(c1,c2): genotype effect: *F_1, 28_* = 49.538, *p* < 0.001; treatment effect: *F_3, 28_* = 1.305, *p* = 0.292; genotype X treatment interaction: *F_3, 28_* = 1.770, *p* = 0.176). As an alternative approach to increase cAMP, forskolin was used to stimulate multiple ADCY. Interestingly, forskolin increased protein translation in both WT and *Fmr1* KO neurons (Figure 5(d1,d2); genotype effect: *F_1, 14_* = 8.788, *p* = 0.009; treatment effect: *F_2, 14_* = 13.539, *p* = 0.001; genotype X treatment interaction: *F_2, 14_* = 0.660, *p* = 0.532).

### 2.7. Effects of PDE2 Inhibition and ADCY Activation on ERK½ and Akt Signaling

Although the function of ADCY1 in regulating ERK½ and Akt signaling has been implicated [10], the overall effect of cAMP through inhibition of PDE2 or non-specific activation of multiple ADCY subtypes has not been investigated. Consistent with that cAMP positively stimulates pERK½, we found that, although there is no effect on total ERK½ (Appendix A), the PDE2 inhibitor BAY607550 upregulated pERK½ in both WT and *Fmr1* KO neurons (Figure 5(e1,e2); WT: *F_3, 28_* = 97.854, *p* < 0.001; *Fmr1* KO: *F_3, 28_* = 33.944, *p* < 0.001). Interestingly, BAY607550 caused a decrease in pAkt in both WT and *Fmr1* KO neurons (Figure 5(e1,e3); WT: *F_3, 28_* = 4.072, *p* = 0.016; *Fmr1* KO: *F_3, 28_* = 4.970, *p* = 0.007) without affecting the level of total Akt (Appendix A).

We found that forskolin causes robust increase of pERK½ (Figure 5(f1,f2); WT: *F_2, 16_* = 36.081, *p* < 0.001; *Fmr1* KO: *F_2, 16_* = 16.800, *p* < 0.001) but decreased pAkt (Figure 5(f1,f3); WT: *F_2, 16_* = 11.182, *p* < 0.001; *Fmr1* KO: *F_2, 16_* = 20.850, *p* < 0.001) in both WT and *Fmr1* KO neurons without affecting total ERK½ (Appendix A) and total Akt level (Appendix A). Together, these results demonstrate that decreasing PDE2A activity or increasing ADCY activity dampens the elevated pAkt but exacerbates the elevated pEK1/2 in FXS mouse neurons.

## 3. Discussion

Our data show that the alteration of cAMP enzymes (i.e., ADCY and PDE) in FXS depends on brain region and developmental stage. Treatment with carbamazepine, which has inhibitory action against ADCY1, is sufficient to correct certain aspects of FXS-associated pathology in a mouse model during adulthood.

As carbamazepine is a pharmacological reagent, it is challenging to definitively pinpoint the mechanism of action underlying the therapeutic effects. Nevertheless, our data show that carbamazepine suppresses both ERK½ and Akt activity. This outcome recapitulates the effect of genetic reduction of ADCY1 on rescuing the elevated ERK½ and Akt signaling in FXS mouse [10]. It is also consistent with the fact that carbamazepine shows pharmacological inhibition activity against ADCY1 [15]. It is important to note that carbamazepine also inhibit ADCY5 and ADCY7 [15]. ADCY7 is not expressed in the brain; ADCY5 is predominately expressed in striatum and nucleus accumbens [26]. Functionally, ADCY5 deficiency leads to increase in ERK½ activity and mGluR5 signaling [26], both of which are associated with cellular pathology in FXS [5,6,27]. ADCY5 deficiency also reduces social interaction [26]. We expect that the carbamazepine effect on ERK/12 and Akt activity in hippocampal neurons and its effect on correcting the social interaction deficits in *Fmr1* KO mice are unlikely mediated through ADCY5 or ADCY7 inhibition.

Along with previous studies, our data show that, while ERK½ activity is positively correlated with cAMP level, Akt activity may be specifically regulated by ADCY1 rather than general cAMP level in neurons. It is evident that specific over-expression of ADCY1 [11] or general increase of cAMP by forskolin (Figure 5(f1–f3)) and PDE2 inhibitors (Figure 5(e1–e3)) can both increase ERK½ activity. However, although genetic reduction of ADCY1 [10] or carbamazepine treatment decreases Akt activity in FXS samples, general increase of cAMP did not cause increase of Akt activity. Forskolin or PDE2 inhibitor rather decreases Akt activity. Thus, targeting ADCY1 may offer a unique opportunity to concurrently suppress the abnormal ERK½ and Akt signaling in FXS.

It is interesting to note that, although ADCY1, ERK½, and Akt are pro-learning molecules and their activity is required for memory formation [18,28,29,30], the elevated activity of these signaling molecules is concurrent with the learning deficits in FXS. It is evident that decrease rather than increase of the activity of ADCY1, ERK½, and Akt shows therapeutic efficacy to correct cognitive impairments in FXS mice [31,32]. Although the mechanism remains elusive, it is possible that concurrent increase of ADCY1, ERK½, and Akt activity along with other pathological changes in FXS impair rather than facilitate learning.

Treatment strategy through an increase of cAMP signaling has been previously proposed [33], although the level of cAMP in crude homogenates has been found to be normal in both human and mouse FXS samples [10,34,35]. As cAMP homoeostasis is maintained by counteracting enzymatic activities from multiple subtypes of ADCY and PDE, identification of alterations in specific cAMP enzyme may help to better understand the disease mechanism. Further, whether alteration of ADCY and PDE in specific brain region, sub-cellular domain, and developmental stage should also be considered. FMRP binds *Adcy1* and *Pde2a* mRNA and the protein levels of ADCY1 and PDE2A are elevated in *Fmr1* KO mice [8,9,10,16]. One complication is that, while elevated ADCY1 is detected in adult FXS mouse samples (Figure 1), alteration of PDE2A is detected in juvenile samples (i.e., at postnatal day 13) [16]. Notably, ADCY1 is enriched at postsynaptic density [36] and PDE2A is substantially expressed at the presynaptic active zone [16]. This indicates an age-dependent and compartmentalized alteration of cAMP enzymes in FXS, suggesting tailored therapeutic approaches. Interestingly, it is found that treatment with PDE2 inhibitor BAY607550 rescues social deficits in infant (postnatal day 10–14) and adolescent (postnatal day 30) *Fmr1* KO animals [25]. Although PDE4D expression has not been compared between normal and FXS samples, chronic treatment with PDE4D negative allosteric modulator (NAM) BPN14770 improves behavioral symptoms associated with social interaction and perseveration [37]. The effect of PDE2 and PDE4 inhibition on cognitive function such as learning and memory has not been determined. In regard to how inhibition of PDE affects the known signaling alteration such as ERK/12 and Akt, our data show that targeting PDE2 dampens the elevated Akt activity but further increases the elevated ERK½ activity in FXS. Moreover, it is worth mentioning that, while ADCYs contribute to cAMP production, PDE2A degrades both cAMP and cGMP. Examination of the therapeutic effect of cGMP may be considered for future investigation.

In summary, our study revealed the efficacy of carbamazepine on correcting the key cellular disease mechanisms including the elevated protein synthesis and ERK½ and Akt signaling. Furthermore, we demonstrated the therapeutic efficacy of carbamazepine to correct adult behavioral abnormalities in a mouse model of FXS. As carbamazepine is an FDA-approved drug, it may offer a practical approach and be clinically repurposed for FXS therapy. It is worth noting that carbamazepine has been mainly used to treat seizure and neuropathic pain. The increased risk of seizure is observed in about 18% of the child FXS patients (from as early as 4 months to 9 years of age) but not in adulthood [38,39]. As a symptom- rather than mechanism-based treatment, anticonvulsants including carbamazepine, valproic acid, lamotrigine, and oxcabazepine have been used to control seizure in FXS children without severe side effects [40]. Nonetheless, our results advocate future tests of clinical efficacy with adult human FXS patients, beyond the anticonvulsant effects in children.

## 4. Materials and Methods

### 4.1. Animals

WT and *Fmr1* KO mice on C57BL6 background (Jackson Laboratory, Stock # 003025) were generated by breeding WT males with *Fmr1* heterozygous females. For all behavioral examinations, 2.5- to 3-month old male mice were used. Animals had free access to food and water and were housed in the Campus Animal Research facility under 12 h light/dark cycle. All procedures (AUF 05-17-086-00, 17 May 2017; PROTO201900290, 29 August 2019; PROTO202000103, 17 August 2020) used in the study were approved by the Institutional Animal Care and Use Committee (IACUC) at Michigan State University on 17 May 2017, 29 August 2019, and 17 August 2020, repectively.

### 4.2. Cell Culture

Neurons [41] and glial cells [42] were cultured from postnatal day 0 WT and *Fmr1* KO mice. Briefly, the cortex and hippocampus were harvested and digested to obtain dissociated neurons and glial cells. Neurons were grown in Neurobasal A supplemented with B27, 0.5 mM glutamine, and antibiotics, and one-third of the growth media was replaced with fresh media once every 3 days. For glia culture, the dissociated cells were maintained in DMEM supplemented with 10%FBS and antibiotics for 9 days and then one-third of the media was replaced with fresh media once every 3 days.

### 4.3. Western Blot Analysis

Samples were first harvested from primary cell cultures on DIV (days in vitro) 14 or from brain tissues of adult (2.5- to 3-month old) mice. After sonication, samples were subjected to 4–20% SDS-PAGE and transferred onto nitrocellulose membranes. The membranes were first incubated with primary antibodies overnight at 4 °C, washed with PBST (PBS with 0.1% Triton-X100), incubated with IRDye 800CW goat anti-rabbit or IRDye 680RD goat anti-mouse secondary antibodies (LI-COR Biosciences, Lincoln, NE, USA, Cat#926-32211 and Cat#926-68070, respectively, 1:5000) for 2 h at room temperature, and washed with PBST. The signals were detected by the Odyssey digital imaging system (LI-COR Biosciences). The primary antibodies were rabbit anti-ADCY1 (Sigma, St. Louis, MO, USA, Cat# SAB4500146, 1:1000), rabbit anti-ERK½ (Cell Signaling, Danvers, MA, USA, Cat#9102, 1:1000), rabbit anti-pERK½ (Cell Signaling, Danvers, MA, USA, Cat#9101, 1:2000), rabbit anti-Akt (Cell Signaling, Danvers, MA, USA, Cat#7272, 1:1000), rabbit anti-pAkt (Cell Signaling, Danvers, MA, USA, Cat#13038, 1:1000), and mouse anti-β-actin (Sigma, St. Louis, MO, USA, Cat#A5541, 1:10,000).

### 4.4. In Vivo Drug Administration

Vehicle (10% DMSO) or 20 mg/kg carbamazepine was i.p. injected 1 h before the behavioral examinations. The examination of passive avoidance memory and object location memory involved training and testing; animals received i.p. injection 1 h before training and were tested 24 h later. For repeated drug administration, the animals received daily injection of vehicle or 20 mg/kg carbamazepine for 8 days, and behavior examinations were performed 1 h after the last injection.

### 4.5. Behavioral Examination

For passive avoidance memory test [19], mouse was placed in the lit side of a passive avoidance chamber (Coulbourn Instruments, Whitehall, PA, USA) and allowed to explore for 1 min, following which the trap door was opened. Once the animal crossed over to the dark side, the trap door was closed and a mild foot shock (0.7 mA for 2 s) was delivered. The mouse remained in the dark side for 30 s before being returned to its home cage. After 24 h, the trained mouse was reintroduced to the lit side and the crossover latency to the dark side was recorded. A maximum of 300 s was used as the cutoff crossover latency value.

For object location memory test [43], mouse was first habituated by exposure to the empty training chamber for 10 min daily for 3 consecutive days. On day 4, the mouse was exposed to the chamber with two objects placed at location A and B. After the 10-min training, the mouse was returned to its home cage. Twenty-four hours later on day 5, the trained mouse was re-introduced to the same chamber with the same objects. One object was placed at the old location A and the other object was place at the new location C. The percentage of time that mouse spent with object at location A and B during training or A and C during testing was scored. Discrimination index was calculated as ((time spend with location C—time spent with location A)/total time spent with objects at both location A and C) during testing.

For open field test [19], mouse was placed in the center of an open field arena and allowed to move freely for 1 h. Ambulatory travel distance in the whole arena and in the center area of the arena as well as the number of entry to the center area were recorded by the Tru Scan Activity System (Coulbourn Instruments, Whitehall, PA, USA).

For social interaction test, we used a 3-chamber paradigm [44] that has been sufficient to detect social deficits in FXS mouse model [45,46]. Briefly, mouse was placed in the center of a 3-chamber social interaction box and allowed to freely explore the entire box for 5 min. Then the mouse was briefly restricted in the center compartment and a stimulus mouse was placed in a wire enclosure on one side and an empty wire enclosure on the other side. The time spent in mouse chamber and in sniffing the stimulus mouse was recorded for 10 min.

For light/dark test [19], mouse was placed in the dark side of a light/dark chamber and habituated for 2 min, after which the trap door was opened. The mouse was allowed to move freely between two chambers. The time spent in the lit chamber and the number of crossing to the lit chamber were recorded for 5 min.

### 4.6. Drug Effects on Protein Synthesis and Neuronal Signaling

Protein synthesis was determined by the SUnSET method, which involves labeling with puromycin [47]. To determine the drug effect, DIV 14 hippocampal neurons were pre-treated with carbamazepine (10, 40, 100 μM), BAY607550 (0.2, 1, 5 μM), or forskolin (1 and 5 μM) for 30 min followed by 30 min incubation with 5 μg/mL puromycin. Samples were harvested at the end of treatment, and protein synthesis level was determined by Western blot using anti-puromycin antibody (KeraFAST, Boston, MA, USA, Cat#EQ0001, 1:1000). The combined intensity of proteins ranging from 20 to 200 kDa was quantified and normalized to the level of β-actin.

To determine the drug effects on neuronal signaling, DIV 14 hippocampal neurons were treated with various concentrations of carbamazepine, BAY607550, or forskolin for 1 h. The treated neurons were harvested. The level of ERK½, pERK½, Akt, pAkt, and β-actin was determined by Western Blot.

### 4.7. Statistical Analysis

Student’s *t*-test was used to analyze data between two groups. Two-way ANOVA followed by Holm–Sidak test was used to analyze the genotype effect and treatment effect. Three-way ANOVA followed by Holm–Sidak test was used to analyze the genotype effect, treatment effect, and location effect for data collected from object location memory test. Three-way repeated measures ANOVA was used to analyze the open field data at various time points. Since the crossover latency during testing in passive avoidance is not normally distributed, the numerical values were converted to categorical data (cut-off value is 300 s) and then analyzed by Fisher’s exact test. All data are presented as average ± SEM. The p value, which is equal to or larger than 0.001, is presented as an actual value. The *p* value, which is less than 0.001, is presented as *p* < 0.001.

## Figures and Tables

**Figure 1 ijms-21-09327-f001:**
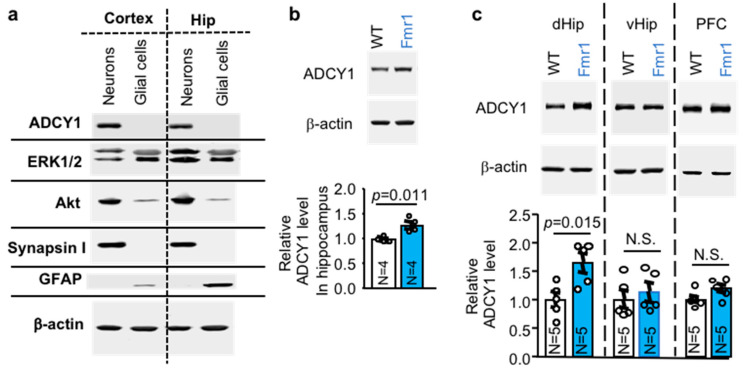
ADCY1 expression in wild type and *Fmr1* knockout samples. (**a**). ADCY1 is expressed in neurons but not in glial cells. Primary cultures of neuron or glial cell were obtained from new born cortex and hippocampus. Cells were harvested on DIV (days in vitro) 14. Expression of ADCY1, ERK½, Akt, synapsin I (a neuronal marker protein), GFAP (a glial cell marker protein), and β-actin in neuronal or glial preparations were determined by Western blot analysis. (**b**). ADCY1 protein expression is elevated in the hippocampus of adult *Fmr1* KO mice. Hippocampal tissues were collected from 3-month old wild type (WT) and *Fmr1* KO (*Fmr1*) mice. The expression of ADCY1 was determined by Western blot and normalized to the level of β-actin. (**c**). ADCY1 expression in the dorsal hippocampus (dHip), ventral hippocampus (vHip), and prefrontal cortex (PFC) of 3-month WT and *Fmr1* KO mice. The level of ADCY1 was normalized with β-actin. Data are presented as average ± SEM. Quantification of ADCY1 level is based on data collected from 4 (in (**b**)) and 5 (in (**c**)) independent biological samples of WT and *Fmr1* KO mice. The *p* values were determined by Student’s *t*-test. N.S.: not significant.

**Figure 2 ijms-21-09327-f002:**
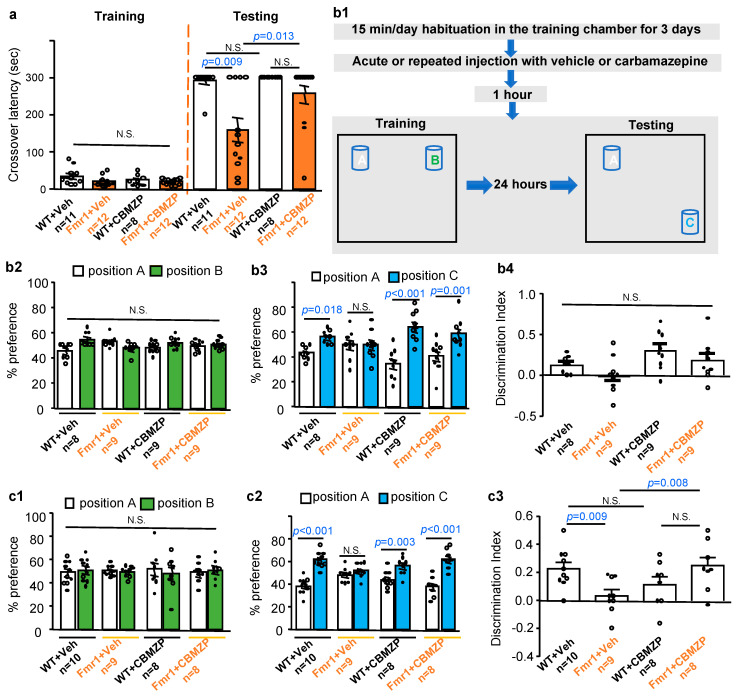
Carbamazepine restores passive avoidance and object location memory in FXS mice. (**a**). Carbamazepine improves passive avoidance memory in FXS mice. 2.5- to 3-month old male wild type (WT) and *Fmr1* KO (Fmr1) mice were first injected with vehicle (Veh) (10% DMSO) or 20 mg/kg carbamazepine (CBMZP). One hour later, they were training by the passive avoidance paradigm and then returned to their home cage. Twenty-four hours later, they were tested. Crossover latency was recorded during training and testing. (**b1**–**b4**,**c1**–**c3**). Carbamazepine improves object location memory in FXS mice. Treatment and behavioral paradigm is shown in (**b1**). 2.5- to 3-month old male mice were trained (**b2**) or (**c1**) and tested (**b3**,**b4**) or (**c2**,**c3**) for object location memory after receiving a single (**b2**) to (**b4**) or repeated (**c1**) to (**c3**) injection with vehicle or carbamazepine. Their preference to the objects during training (**b2**,**c1**) and testing (**b3**,**c2**) is shown. The object discrimination index during testing is shown in (**b4**,**c3**). All data are presented as average ± SEM. Two-way ANOVA followed by post hoc analysis is use for the passive avoidance training data (**a**) and the object discrimination index data (**b4**,**c3**). Fisher exact is used for data collected during passive avoidance testing. Three-way ANOVA followed by post hoc analysis is used to analyze the object location training/testing data. N.S.: not significant.

**Figure 3 ijms-21-09327-f003:**
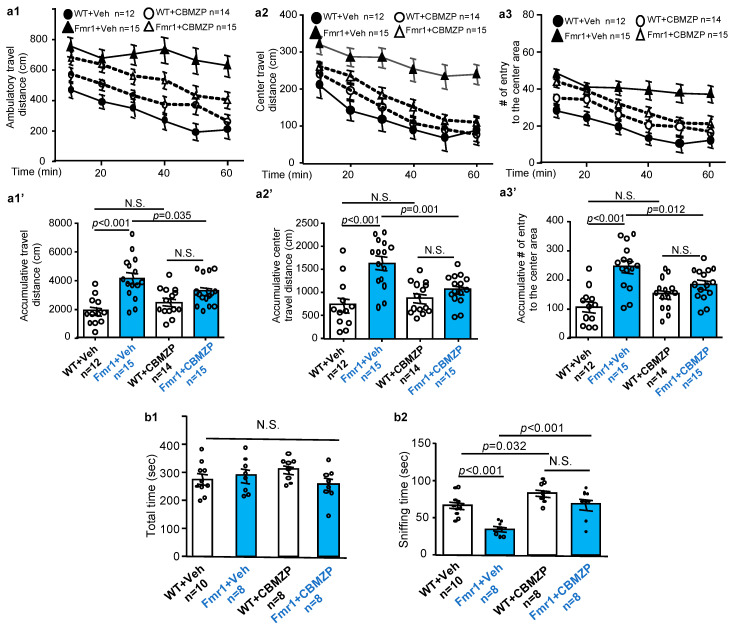
Carbamazepine corrects hyper locomotion and social deficits in FXS mice. 2.5- to 3-month old male wild type (WT) and *Fmr1* KO mice received a single (**a1**–**a3**) or repeated (once/day for 8 days) (**b1**,**b2**) injection with vehicle (Veh) vehicle (Veh) (10% DMSO) or 20 mg/kg carbamazepine (CBMZP). (**a1**–**a3**). 60 min after the injection, mice were subjected to a 1-h open filed test. The ambulatory travel distance in the whole open field arena and center area for each of the 10-min bin is presented in (**a1**,**a2**), respectively. Number of entry to the center area for each of the 10-min bin is presented in (**a3**). The corresponding cumulative activity during the whole 1-h testing is presented in (**a1′**–**a3′**). (**b1**,**b2**). 60 min after the last daily injection, mice were subjected to a 3-chamber social interaction test. Total time spent in the stimulus mouse chamber is presented in (**b1**). Direct social interaction between the test mouse and the stimulus mouse was determined by the amount of sniffing time and presented in (**b2**). Data are presented as average ± SEM. Three-way (**a1**–**a3**) or two-way ANOVA (**a1′**–**a3′**,**b1**,**b2**) were used for data analysis. Difference between two groups was determined by post hoc analysis following ANOVA test. N.S.: not significant.

**Figure 4 ijms-21-09327-f004:**
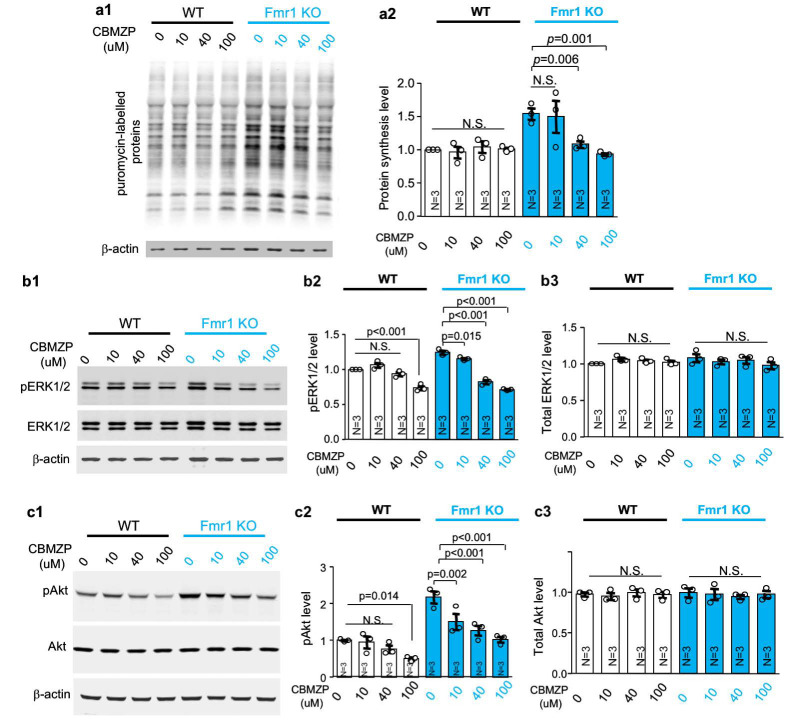
Carbamazepine corrects the aberrantly elevated protein synthesis and neuronal signaling in FXS mouse neurons. DIV (days in vitro) 14 neurons cultured from hippocampus of wild type (WT) and *Fmr1* KO mice were treated with various concentrations of carbamazepine (CBMZP, as indicated). Samples were harvested 60 min after treatment. (**a1**,**a2**). Protein synthesis was labelled with puromycin and determined by Western blot with antibody against puromycin. The level of puromycin-labelled protein synthesis was normalized to β-actin. (**b1**–**b3**,**c1**–**c3**). The level of pERK½ and total ERK½ (**b1**–**b3**) and pAkt and total Akt (**c1**–**c3**) was determined by Western blot. The level of pERK½ and pAkt was normalized to total ERK½ and Akt, respectively. The level of total ERK½ and Akt was normalized to β-actin. Representative results are presented in (**a1**,**b1**,**c1**). Quantification is presented as average ± SEM in (**a2**,**b2**,**b3**,**c2**,**c3**). The N number indicated in the figure is the number of experiments performed with independent neuronal cultures. The *p* values were determined by two-way ANOVA followed by post hoc analysis. N.S.: not significant.

**Figure 5 ijms-21-09327-f005:**
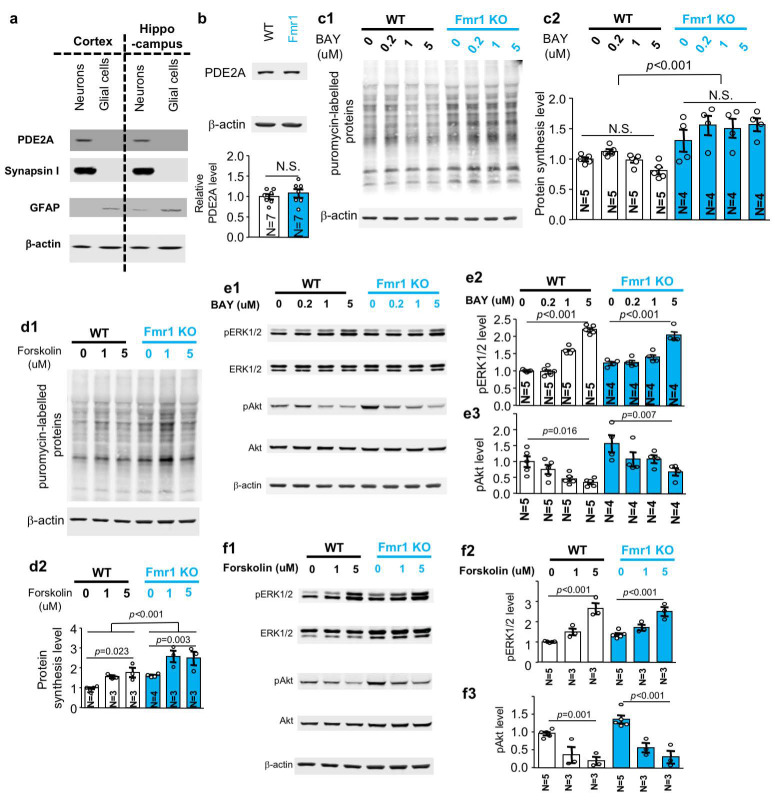
Effects of pharmacological inhibition of PDE2 and general activation of ADCY on protein synthesis and ERK½ and Akt activity. (**a**). PDE2A is expressed in neurons but not in glial cells. As described in Figure 1a, the expression of PDE2A, synapsin I, GFAP, and β-actin in neuronal or glial preparations were determined by Western blot analysis. (**b**). PDE2A protein is expressed at comparable level in the hippocampus of 3-month old wild type (WT) and *Fmr1* KO (*Fmr1*) mice. The level of PDE2A was normalized with β-actin. (**c1**,**c2**) to (**f1**–**f3**), DIV 14 neurons from WT and *Fmr1* KO mice were treated with PDE2 inhibitor BAY607550 (BAY) (**c1**,**c2**,**e1**–**e3**), and ADCY stimulator forskolin (**d1**,**d2**,**f1**–**f3**) at various concentrations (as indicated) for 1 h. (**c1**,**c2**,**d1**,**d2**). Protein synthesis was labelled with puromycin and determined by Western blot with antibody against puromycin. Representative results are shown in (**c1**,**d1**). Quantification (normalized with β-actin) is shown in (**c2**,**d2**). (**e1**–**e3**,**f1**–**f3**). The level of total ERK½, pERK½, total Akt, pAkt, and β-actin was determined by Western blot. The level of pERK½ and pAkt was normalized to total ERK½ and Akt, respectively. Representative results are shown in (**e1**,**f1**). Quantification is presented as average ± SEM. Quantification for pERK½ is presented in (**e2**,**f2**). Quantification for pAkt is presented in (**e3**,**f3**). The N number indicated in the figure is the number of experiments performed with independent neuronal cultures. The *p* values were determined by Student’s *t*-test (**b**), two-way (**c1**,**c2**) to (**f1**–**f3**) ANOVA followed by *post-hoc* analysis. N.S.: not significant.

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
