# Peer review of "Carbamazepine Restores Neuronal Signaling, Protein Synthesis, and Cognitive Function in a Mouse Model of Fragile X Syndrome"

_ijms, 2020, doi:10.3390/ijms21239327_

Round 1

Reviewer 1 Report

Summary:

Fragile X syndrome (FXS) is a leading genetic cause of intellectual disability and doesn’t have a treatment yet. This manuscript focuses on elucidating the effect of carbamazepine, an FDA-approved drug for mainly treating seizure and neuropathy pain. In this study, authors showed that carbamazepine rescues cognitive impairments, hyperlocomotion and social deficits in Fmr1 KO mice, a mouse model for FXS. Authors also claim that carbamazepine dampens the elevated ERK1/2 and Akt signaling as well as protein synthesis in Fmr1 KO mice, implicating the potential use of this drug for treating FXS. Given the known counteracting enzymatic activity on cAMP production by ADCY and PDE2, inhibiting the elevated levels of both FMRP downstream proteins could lead a complication by using carbamazepine, which authors try to understand deeper about in this manuscript. Authors demonstrate that carbamazepine-mediated downregulation of both signaling have different effect, which can be a critical information about how two different pathways contribute to cAMP homeostasis in FXS. Overall, although the study is novel and it provides a deeper understanding on molecular mechanisms underlying FXS, there are a few points that authors need to address as mentioned below:

<Major points>

Figure 1: GFAP and ADCY1 Western blotting images are not seen in Figure 1a. Basically bands are missing in the representative images in Figure 1a. Also, can authors indicate among how many experiments performed the representative images of Western blotting are from in Figure 1?

Figure 3: It looks like authors observed the increase of sniffing time after a repeated (8 days) treatment of carbamazepine in WT mice according to Figure b2. Given that a single injection doesn’t have that effect in WT mice, could this be only seen when treated repeatedly? Given that this can be potentially an unwanted effect of using carbamazepine, can authors at least mention or discuss about this in the result section or discussion section?

Figure 4: Can authors indicate among how many experiments performed the representative images of Western blotting are from in Figure 4? Have authors measured ERK1/2 (total and phosphorylated) and Akt (total and phosphorylated) in vivo by immunostaining or Western blotting similar to what was studied in reference #10? Authors could measure in vivo levels from the same mice that went through behavior tests from Figure 2 and 3. If not, authors at least need to explain why authors focused only in primary cultured neurons in the result or discussion section since in vivo treatment output and culture condition can differ significantly.

Figure 5: Can authors indicate among how many experiments performed the representative images of Western blotting are from in Figure 5?

<Minor points>

  1. Overall, authors need to review the entire manuscript and check typos.

- page 2, line 62, typo of ‘inhibition’

- page 2, line 69, typo of ‘Fmr1’

- page 6, line 140, typo of ‘Fmr1’

  1. Authors indicated that some of P values are 0.000 both in the text and Figures. Did authors mean P<0.001? It is not clear. Please define in the statistical analysis section or legends or in the main result section to make it clearer.

  1. ‘Animals’ section in ‘Materials and Methods’ should have more detailed information (strain, company, catalog number).

Author Response

We are submitting a revision of manuscript ijms-994183, which has been reviewed and recommended for a major revision. We appreciate the reviewers’ insightful comments, which help us to improve the manuscript. All changes are in the red-colored text within the revised manuscript. We provide point-by-point responses.

Review 1

Comment 1. In Figure 1, GFAP and ADCY1 Western blotting images are not seen in Figure 1a. Basically bands are missing in the representative images in Figure 1a. Also, can authors indicate among how many experiments performed the representative images of Western blotting are from in Figure 1?

      We are sorry that the information was lost. The representative images, which were present in the uploaded figure files but got lost when they were inserted into the text, are now inserted correctly in the revised manuscript file. The number of experiments is presented in the figure and described in the figure legend of the revised manuscript.

      The following description is provided in the revision. Quantification of ADCY1 level is based on data collected from 4 (in b) and 5 (in c) independent biological samples of WT and Fmr1 KO mice (see lines 93 and 94).

Comment 2. In Figure 3, it looks like authors observed the increase of sniffing time after a repeated (8 days) treatment of carbamazepine in WT mice according to Figure b2. Given that a single injection doesn’t have that effect in WT mice, could this be only seen when treated repeatedly? Given that this can be potentially an unwanted effect of using carbamazepine, can authors at least mention or discuss about this in the result section or discussion section?

      We appreciate the comment. In the revised manuscript, we point out the effects of repeated carbamazepine in both WT and Fmr1 KO mice.

      The following description is provided in the revision. It is interesting to note that repeated carbamazepine also increased social interaction in WT mice (Fig. 3b2), implicating that social behavior in WT mice can also be improved. Importantly, following repeated carbamazepine administration, WT and Fmr1 KO mice showed comparable social interaction (Fig. 3b2; genotype effect: F1, 30=19.521, p=0.000; treatment effect: F1, 30=22.184, p=0.000; genotype X treatment interaction: F1, 30=2.620, p=0.116) (see line 163 to 165).

Comment 3. Can authors indicate among how many experiments performed the representative images of Western blotting are from in Figure 4? Have authors measured ERK1/2 (total and phosphorylated) and Akt (total and phosphorylated) in vivo by immunostaining or Western blotting similar to what was studied in reference #10? Authors could measure in vivo levels from the same mice that went through behavior tests from Figure 2 and 3. If not, authors at least need to explain why authors focused only in primary cultured neurons in the result or discussion section since in vivo treatment output and culture condition can differ significantly.

      We appreciate the insightful comment. The N number indicated in the figure is the number of experiments performed with independent neuronal cultures (see lines 205 and 206). The alteration of ERK1/2 and Akt activity, which is considered as potential cellular abnormalities associated with FXS, has been  previously reported in brain tissues. As the brain tissue consists of both neuron and non-neuronal cell types (e.g., glial cells), it is not clear whether the abnormal ERK1/2 and Akt is neuron specific or due to changes in non-neuronal cells. By using primary cultures enriched for neurons, our results identify neuron-specific exaggeration of ERK1/2 and Akt activity. We further found that carbamazepine dampens these FXS-associated abnormalities in neurons. 

      We provide the following description in the revision. The elevated ERK1/2 and Akt activity, which may contribute to the exaggerated protein synthesis in FXS neurons, has been previously observed in brain tissues of Fmr1 KO mice. As the brain tissue consists of both neuron and non-neuronal cell types (e.g., glial cells), it is not clear whether the abnormal activity of ERK1/2 and Akt is neuron specific or due to changes in non-neuronal cells. Here, we used neuron-enriched primary cultures to determine whether there is neuron specific alteration of ERK1/2 and Akt and examine the therapeutic effect of carbamazepine (see line 210 to 215).

Comment 4.  For Figure 5, can authors indicate among how many experiments performed the representative images of Western blotting are from in Figure 5?

      The number of experiments is presented in the figure and described in the figure legend of the revised manuscript. The N number indicated in the figure is the number of experiments performed with independent neuronal cultures (see lines 282 and 283).  

Comment 5. Overall, authors need to review the entire manuscript and check typos.

      We run spelling and grammar check with Microsoft Word and Grammarly. The changes are in red (see lines 39, 48, 62, 66, 69, 80, 97, 143, 351, 410).

Comment 6. Authors indicated that some of P values are 0.000 both in the text and Figures. Did authors mean P<0.001? It is not clear. Please define in the statistical analysis section or legends or in the main result section to make it clearer.

      We provide the following description in the “Materials and Methods” section of the revision. The p value, which is equal to or larger than 0.001, is presented as an actual value. The p value, which is less than 0.001, is presented as p<0.001 (see lines 444 and 445). All “p=0.000” is changed to “p<0.001” in the revised manuscript (see lines 116, 123, 148, 150, 151, 166, 167, 218, 221, 222, 246, 258, 262, 263,  457, 468, 469).

Comment 7. ‘Animals’ section in ‘Materials and Methods’ should have more detailed information (strain, company, catalog number).

      We update the information. WT and Fmr1 KO mice on C57BL6 background (Jackson Laboratory, Stock # 003025) were generated by breeding WT males with Fmr1 heterozygous females (see line 356).

Reviewer 2 Report

The study uses FMR1 KO mouse model to study effect of Carbamazepine, an anticonvulsant drug, on cognitive and social/behavioral deficiencies of Fmr1 KO mice model. Mechanistically they show Carbamazepine decreases protein synthesis, pERK and pAKT in neuronal cells.

Major:

  • In Figure-1 shows increased expression of ADCY1 dorsal hippocampus of Fmr1 KO mice. It would be good to show, effect of Carbamazepine on ADCY1 expression in dorsal hippocampus region.
  • Figure-4 shows rescue of elevated protein synthesis and dose dependent decrease in pERK and pAKT level in isolated neuron. How about protein synthesis, pERK and pAKT level in Different regions of mice brain in ip dosed mice?
  • It would be also interesting to see effect of Carbamazepine on other related dysregulated markers in Fmr1KO mice For eg. S6K, MMP9

Minor: 

  • Few spelling and grammatical mistakes. page 39: grammatical- "enhanced" pg 336: spelling of anticonvulsant;
  • Figure legend should include details age of mice, dose and duration of Carbamazepine

Author Response

We are submitting a revision of manuscript ijms-994183, which has been reviewed and recommended for a major revision. We appreciate the reviewers’ insightful comments, which help us to improve the manuscript. All changes are in the red-colored text within the revised manuscript. We provide point-by-point responses.

Comment 1. Figure-1 shows increased expression of ADCY1 dorsal hippocampus of Fmr1 KO mice. It would be good to show, effect of Carbamazepine on ADCY1 expression in dorsal hippocampus region.

      We respect this comment. We would like to point out that carbamazepine has been shown to inhibit ADCY1 activity rather than its expression level. We appreciate that the reviewer would agree that administration of carbamazepine is unlikely to change the level of ADCY1.

Comment 2. Figure-4 shows rescue of elevated protein synthesis and dose dependent decrease in pERK and pAKT level in isolated neuron. How about protein synthesis, pERK and pAKT level in Different regions of mice brain in ip dosed mice?

      We appreciate the insightful comment. The alteration of overall protein synthesis, which requires in vitro metabolic labelling, has been mostly detected with cell culture or ex vivo brain slices. There is technical limitation to determine in vivo translation.

      The alteration of ERK1/2 and Akt activity, which is considered as potential cellular abnormalities associated with FXS, has been  previously reported in brain tissues. As the brain tissue consists of both neuron and non-neuronal cell types (e.g., glial cells), it is not clear whether the abnormal ERK1/2 and Akt is neuron specific or due to changes in non-neuronal cells. By using primary cultures enriched for neurons, our results identify neuron-specific exaggeration of ERK1/2 and Akt activity. We further found that carbamazepine dampens these FXS-associated abnormalities in neurons. 

      We provide the following description in the revision. The elevated ERK1/2 and Akt activity, which may contribute to the exaggerated protein synthesis in FXS neurons, has been previously observed in brain tissues of Fmr1 KO mice. As the brain tissue consists of both neuron and non-neuronal cell types (e.g., glial cells), it is not clear whether the abnormal activity of ERK1/2 and Akt is neuron specific or due to changes in non-neuronal cells. Here, we used neuron-enriched primary cultures to determine whether there is neuron specific alteration of ERK1/2 and Akt and examine the therapeutic effect of carbamazepine (see line 210 to 215).

Comment 3. It would be also interesting to see effect of Carbamazepine on other related dysregulated markers in Fmr1KO mice For eg. S6K, MMP9.

      We respect this insightful comment. Although how FMRP regulates translation is still not clear, previous studies suggest two potential mechanisms. On the one hand, lack of FMRP may alter neuronal signaling (such as abnormality of cAMP and ERK1/2 activity), which in turn affects the overall activity of the translation machinery and causes a non-specific and general increase of protein synthesis. On the other hand, FMRP may directly interact with its target mRNAs (such as S6K and MMP9) and suppress the translation of these specific targets. Our study focuses on whether attenuation of neuronal signaling can correct the overall protein synthesis in Fmr1 KO neurons. We appreciate that the reviewer would agree that whether correction of the abnormal neuronal signaling can also affect the translation of specific FMRP targets is an interesting but different topic, which may be investigated in future studies.

Comment 4. Few spelling and grammatical mistakes. page 39: grammatical- "enhanced" pg 336: spelling of anticonvulsant.

      We run spelling and grammar check with Microsoft Word and Grammarly. The changes are in red (see lines 39, 48, 62, 66, 69, 80, 97, 143, 351, 410).

Comment 5. Figure legend should include details age of mice, dose and duration of Carbamazepine.

      The information is added in figure legend (see lines 128, 129, 130, 133, 175, 176, 177, 178, 182, 450, 451, 452, 463, 464, 465).

We appreciate the expertise and evaluation from the reviewers. We expect that, by following the reviewers’ suggestion, our manuscript is improved.

Round 2

Reviewer 2 Report

The concerns raised has been mostly addressed by the authors with appropriate changes in the main text.